



# Evaluation of Sub-Monthly Oceanographic Signal in GRACE "Daily" Swath Series Using Altimetry

Jennifer A. Bonin[1] and Himanshu Save[2]

[1]College of Marine Science, University of South Florida, St. Petersburg, FL, 33701, USA
[2]Center for Space Research, University of Texas at Austin, Austin, TX, 78759, USA

*Correspondence to*: Jennifer A. Bonin (jbonin@mail.usf.edu)

**Abstract.** Bottom pressure estimates from three different GRACE gravity series (an experimental CSR swath series, ITSG2016, and ITSG2018) and two global ocean models (OMCT and MPIOM) are compared to Jason altimetry sea level
anomaly estimates, in order to determine the accuracy of the GRACE and model data at sub-monthly time scales. We find that the GRACE series are capable of explaining 25-75% of the sub-monthly altimetric variability over most of those ocean regions which have high signal strength. All three GRACE series explain more of the sub-monthly variability than the de-aliasing products they were created with do. Upon examination over finer frequency bands, the GRACE series prove superior at explaining that altimetric signal for signals with periods as short as 10 days.

## 1 Introduction

Many Earth-observing satellite missions utilize high-frequency oceanographic models to prevent quick-moving geophysical signals from aliasing into longer-period errors under the effect of a relatively long orbital repeat pattern. Both the Gravity Recovery and Climate Experiment (GRACE) mission (Tapley et al., 2004; Wouters et al., 2014) and the Jason family of radar
altimeters (Lambin et al., 2010; Ménard et al., 2003) use such ocean de-aliasing models. For such de-aliasing to be successful, the models must naturally be close to reality, else the errors in using them might be as large as not de-aliasing at all. Over the past two decades, the sub-monthly global de-aliasing models have improved substantially. Yet they remain imperfect, particularly in hard-to-observe areas such as the distant and deep southern ocean. In this work, we utilize Jason satellite altimetry data to demonstrate that sub-monthly GRACE data can improve upon the existing GRACE de-aliasing model in
several high-signal regions. We then attempt to measure over which frequency bands GRACE is more like altimetry than the existing model is, and thus where and when it might add value over the current de-aliasing model.

The work in this paper was predicated by Bonin and Chambers (2011), called BC11 henceforth. BC11 used older altimetry, model, and GRACE data, but was still able to demonstrate that the then-modern daily ITG-GRACE2010 series could explain





10-30% more of altimetry's variance than the de-aliasing model of the day could, across large bands of the southern ocean and
northern Pacific. The theory which both that paper and this one are based upon is straight-forward. If GRACE is observing
real sub-monthly ocean bottom pressure signals, then the variability of the difference between GRACE and altimetry (along
the altimetry groundtracks) ought to be smaller than the variability within altimetry alone. Or, put in a more mathematical
form, if GRACE is seeing real signal, then:


$$\text{var(Altimetry – GRACE)} < \text{var(Altimetry)} \qquad (1)$$

In this case, the percent of altimetry's variance explained will be greater than zero. Similarly, if GRACE is more accurate than
the de-aliasing model, then a double-difference of the variances will show that:


$$\text{var(Altimetry – GRACE)} < \text{var(Altimetry - Model)} \qquad (2)$$

Or in terms of relative percent variance explained:


$$\text{pervar(Alt. explained by GRACE) - pervar(Alt. explained by Model)} > 0 \qquad (3)$$

We do not expect this method of comparison to work well in places with large short-wavelength signals, which neither GRACE
nor the models resolve well, nor where altimetry is measuring ocean height changes caused primarily by temperature or salinity
differences, which GRACE cannot see at all. However, as we determined in BC11, this restricts us only from regions with
strong currents, such as along the western boundaries of the continents and the center of the Antarctic Circumpolar Current,
leaving most of the ocean accessible to this analysis.

Our ultimate goals in this work are two-fold. First, we wish to demonstrate conclusively that the modern "daily" GRACE
series we consider are truly measuring real sub-monthly signal, and that not all of that signal is coming from their apriori ocean
de-aliasing models. Second, unlike in BC11, we wish to place more specific bounds on which frequency bands GRACE is
able to measure better than its de-aliasing model does. The ultimate goal which we aspire to in future work is to create a
combined sub-monthly ocean de-aliasing model, where the highest frequencies and shortest wavelengths remain purely model-
based, but the longer frequencies and larger spatial scales are blended with those pieces of information from GRACE which
are statistically likely to be more accurate.




## 2 Methods

The procedures followed in this work are largely those laid out in BC11. In broad form:

1.)  Ocean bottom pressure data from GRACE and sea-level anomaly data from Jason-1 and Jason-2 are collected and processed in the standard manner. Bottom pressure estimates from the ocean de-aliasing models used with GRACE are also prepared. Details on this step are given in the following section.

         2.)  Since the altimetry data exists only in those places and times which Jason overflies, the GRACE and model data are
masked to cover only those same groundtracks. On days where either GRACE or altimetry data does not exist, no data of any type is utilized. All data types are averaged into daily files at 3° resolution. The altimetry and model data are smoothed using a 300km Gaussian spatial filter, in order to better match them with the native GRACE resolution. This also reduces the impacts of eddies and other short-wavelength phenomena.

3.)  All data types are band-pass filtered, to recover information within three frequency regimes: signals with periods shorter than 10 days, between 10 and 20 days, and between 20 and 30 days. Gaussian temporal windows are used to segment these regimes so as to avoid incorporating unwanted frequencies due to side-lobes. In addition to these three frequency bands, the sub-monthly variability is considered as a whole, as in BC11.

4.)  The differences between GRACE and altimetry, and also between the models and altimetry, are computed over each frequency band. The variances of each differenced series are measured. The percentage of altimetry's variance which is explained by GRACE or the models is computed for each frequency band. A double-difference estimate of this percent variance explained (Eq. 3) demonstrates visually and statistically which GRACE or model estimate explains more of the altimetry signal in the various areas and frequency bands.

In addition to the updated GRACE, model, and altimetry data series, the main difference between this work's techniques and those in BC11 is the use of more frequency bands, so that the type of sub-monthly variability can be more closely analyzed. The 10-day altimetry orbital repeat period necessitates the use of larger (3°) grid cells, such that sufficient data is retained within most grids over the shortest frequency band.

## 3 Data Products

### 3.1 Jason Altimetry

The estimates of sea level anomaly considered here come from the most modern Jason-1 and Jason-2 Geophysical Data Records (GDR) at this time. We use Jason-1 GDR version E from 2002-2008, and Jason-2 GDR version D from 2009-2016. These products are freely available (ftp://podaac.jpl.nasa.gov/allData/jason1/L2/gdr_netcdf_e/



and ftp://ftp.nodc.noaa.gov/pub/data.nodc/jason2/gdr/gdr/). All standard corrections are applied, including the inverted barometer correction. However, the non-tidal ocean de-aliasing model contained within the dynamic atmospheric correction
(DAC) is *not* removed, as that is what we are attempting to observe.

While computing the sea level anomaly values, we stumbled across one surprise difficulty which we would like to make particular note of, for the benefit of other Jason data users. When we went to reference the Jason-2 data to the Jason-1 data, we determined that there is a large, geographically-correlated bias between the most modern data releases of the two missions
(Fig. 1a). To provide a historical backdrop, Jason-1 produced solo data from January 2002 through July 2008, at which point it was joined by Jason-2. The two satellites flew in the same orbit, one shortly behind the other, until the start of 2009, at which point Jason-1 was moved into an interleaved complementary orbit. The last of the Jason-1 data was captured in March 2012, after which Jason-2 flew alone for four years. In February of 2016 (after GRACE finished) Jason-3 was added in an identical orbit for eight months, until Jason-2 was moved to the complimentary orbit through its demise in May 2017. The
common way of aligning the two missions is to take the data during the overlap periods and compute a bias from that. Historically these mission biases have pointed out several complex error types, but those are now understood, so that the Jason-1 GDR-D to Jason-2 GDR-D biases were constant across the ocean. When we compute the Jason-2 GDR-D to Jason-3 GDR-D biases (Fig. 1b), that is still what we see.

Thus our surprise upon finding a large, non-uniform Jason-1 GDR-E to Jason-2 GDR-D bias! Thanks to the experienced aid of Don Chambers, we have determined that this bias pattern comes from a change in the mean sea surface (MSS) model used between the D and E versions of the GDR output. In version D, a 16-year mean dynamic topography was used to create the MSS correction; in version E, a 20-year mean was used (Eumetsat et al., 2016). Not only are there small differences because of the change in time-period used to create the mean, but also, the temporal zero-point of the E version occurs 4 years after the
D version does, resulting in the large bias seen.

To correct for this properly, one would ideally need to reprocess both (or all three) sets of Jason data with a consistent MSS model. This being a lengthy process, however, we used the following approximate technique instead. We computed the average overlap bias (Fig. 1a) along the ground-tracks, then smoothed it with a 100km Gaussian smoother to remove very
short-wavelength features, and used the value of that at each point as the Jason-2 bias correction. We note that it is very important to compute the value at each point, as there would be a ±4 cm spread in water heights otherwise, depending on where in the ocean you are looking. Using the mean bias over this map would result in a constant -0.785 cm correction, which would not be accurate in most places. (As a comparison, we found the Jason-2D to Jason-3D bias to be a larger but very stable -2.871 cm.)




## 3.2 GRACE Gravity

While most Gravity Recovery and Climate Experiment (GRACE) series have been released at monthly intervals in spherical harmonic form with a maximum spatial resolution of 300-500 km, over the past decade there have been several attempts at
resolving the GRACE data into higher frequencies in both space and time. We consider three different modern GRACE "daily" series here.

The main GRACE series, called "CSR swath" throughout, is an experimental mascon (mass-concentration) product created at the Center for Space Research (CSR) at the University of Texas at Austin (Save et al., 2018). The CSR GRACE swath
solutions used here are an extension of the CSR RL05 GRACE monthly mascon solutions (Save et al., 2016) but computed daily for those mascons observed by a particular day's ground track. Though the full extent of this data is not currently available for public release, a subset consisting of only the ocean grid cells between 66°S and 66°N has been placed at <A website will be provided here before publication>. This will allow for replication of this paper's techniques, as desired.

The CSR swath series is a near-daily solution, with the world divided into 40962 equal-area geodesic mascon blocks with an average distance of 120 km across. Equivalent water layer (cm) anomalies are computed each orbital pass by estimating the mass change observed in a narrow swath around the GRACE groundtrack. These passes are then accumulated over consecutive days to give an estimate of the global mass anomaly at shorter temporal resolution. The mass estimate of each mascon is only updated when GRACE satellites overfly the mascons within 220 km from the center of the mascon. The mascons at high
latitudes are observed every 1-2 days while a few mascons near the equator are observed only once every 4-5 days depending on how the groundtracks lay out over time. Typical ocean mascons at mid-latitudes are observed once every 2-4 days. A statistical combination of older data and newer neighboring data fills in the gaps. The global mascon solutions and regularization are purely driven by GRACE without any influence from external models. The time-variable regularization process used does not bias or attenuate future regional signals based on statistics from models or past GRACE months, but is
designed to encourage no land/ocean correlation in order to reduce leakage.

During processing, the sub-monthly release-5 GRACE de-aliasing model ("AOD5", see next section) was removed as one of the standard apriori background models. The GRACE swath series thus estimates the ocean mass change relative to this AOD5 model. We have restored the model, such that the results shown here are the updated combination. The CSR swath series has
also had a GIA model removed (A et al., 2013), though at sub-monthly scale that is noncritical. No geocenter information is included, because GRACE cannot measure it. We chose not to insert an external geocenter estimate, since geocenter series with daily output are rare, and the accuracy on the weekly and sub-weekly scale is highly uncertain and likely to be poor [for example: *Männel and Rothacher*, 2017]. Similarly, the GRACE $C_{20}$ signal not replaced and thus assumed to be correct.





The result is a smoothly-varying series with the ability to pinpoint signals very well spatially (Fig. 2a). The CSR swath technique (like most other mascon methods) has the ability to separate ocean and land signals reasonably well, thus decreasing leakage from land and ice-covered areas into the ocean. The swath series also tends not to show the classic north-south "stripe" errors which are customary with GRACE spherical harmonic solutions. The practical combination of temporal and spatial accuracy can be examined by looking at the average signal at each time over relatively small areas of the world. Consider the

average across the Argentine ocean basin, off the coast of Brazil (Fig. 2b). While it is likely that many of the spikes seen there are error-driven, there are significant sub-monthly-scale features picked up which are hoped to be real.

We also consider two related secondary GRACE series, ITSG2016 and ITSG2018, created by the Technische Universität Graz (TU Graz). The older of the two, ITSG2016 (Norbert, 2016), can be downloaded at

https://www.tugraz.at/institutes/ifg/downloads/gravity-field-models/itsg-grace2016/, while the new and still somewhat experimental ITSG2018 (Norbert, 2018) can be found at https://www.tugraz.at/institute/ifg/downloads/gravity-field-models/itsg-grace2018/. Both series are created in spherical harmonic form to maximum degree/order 40. Daily resolution is achieved via Kalman smoothing, such that it is stabilized based on geophysical models. ITSG data exists even when GRACE does not augment it, but it tends toward the models used, so we omit all days when the CSR swath data denotes a gap. The

two ITSG versions differ in the background models used and the details in how instrumental processing steps were handled (see the above websites for details). Significantly, ITSG2016 uses the release 5 ocean de-aliasing model (AOD5), while ITSG2018 uses the newer release 6 version (AOD6, see next section).

**3.3 De-aliasing Ocean Models**

It is critically important for the production of monthly GRACE gravity products that sub-monthly changes within the ocean which could cause gravitational anomalies are estimated and removed from the GRACE data before processing. Doing so prevents sub-monthly signals from aliasing incorrectly into the monthly estimates. Even the higher-frequency GRACE series

remove these modelled estimates from their input before processing the gravity fields, and then restore them later. This means the CSR swath and ITSG solutions considered here actually solve for the residual gravitational signal between the reality and the ocean model. In this paper, we look for evidence that GRACE can see not merely the reproduced apriori model, but additional high-frequency ocean signal above and beyond that.

We will consider the oceanic bottom pressure signals ("GAD" products) of the two most recent GRACE Atmosphere and Ocean De-aliasing (AOD) models, AOD release 5 (Dobslaw et al., 2013; Flechtner et al., 2014) and AOD release 6 (Dobslaw

et al., 2017). These ocean bottom pressure products are the output of high-resolution baroclinic ocean models and do not measure the influence of tides.

AOD5's ocean bottom pressure estimates come from the Ocean Model for Circulation and Tides (OMCT) (Thomas, 2002), a baroclinic ocean model which estimates the state along 1° horizontal grids with 20 vertical layers and time-steps of 20 minutes. OMCT is forced every six hours with the European Centre for Medium-Range Weather Forecasts (ECMWF) atmospheric pressures, wind stresses, temperatures, and freshwater fluxes. Its bottom pressure output is made available for GRACE processing every six hours with spatial resolution given by maximum spherical harmonic degree/order 100 (~200km). The

CSR swath and ITSG2016 GRACE series were both made using AOD5.

The bottom pressure estimates of AOD6, the very newest GRACE generation's de-aliasing product, are instead based on the Max Planck Institute for Meteorology Ocean Model (MPIOM) (Jungclaus et al., 2013), a global general circulation ocean model which is a cousin to OMCT, not a direct descendant. The innate spatial resolution of this model is 1° along the horizontal

with 40 vertical layers, and time-steps are given as 90 minutes. It is similarly forced by ECMWF input data, but outputs every 3 hours rather than every 6 and to a maximum spherical harmonic degree/order of 180 (~111km).

As a comparison, we also briefly consider the high-frequency ocean component of the de-aliasing product used with the Jason altimetry data. A run of the barotropic, non-linear, finite-element model Mog2D (Lynch and Gray, 1979) is used for this,

representing the impacts of the ECMWF winds and pressure fields at frequencies with periods below 20 days (note: this is different than the ~30-day months used elsewhere in this paper). The full Dynamic Atmospheric Correction (DAC) product is unfortunately released as a combination of Mog2D with the larger inverted barometer model, which contains significant sub-20-day signal as well and thus cannot be easily separated (https://www.aviso.altimetry.fr/en/data/products/auxiliary-products/atmospheric-corrections/description-atmospheric-corrections.html). However, a separated subset of the high-

frequency Mog2D altimetry de-aliasing ocean model is available, only along the Jason groundtracks, within the GDR records themselves, which is what we use here.

## 4  Sub-monthly Results

As an example of the statistics we will be computing, we show the point-by-point standard deviation of the full sub-monthly signal from CSR swath, altimetry, and the difference of CSRswath-altimetry in Fig. 3. Notice how the altimetry sea level anomaly product shows a very large (>5 cm water height) variability within the major current systems. GRACE does not see these signals, either because they are short-wavelength signals (for example, eddies) beneath the ~300-km resolution of



GRACE, or because they are sea surface height changes which do not correspond to a mass change (for example, those caused
by a change in temperature, not pressure). Additionally, there are several areas of high, short-wavelength ocean activity, such
as the Argentine gyre southeast of Brazil, where GRACE registers only a reduced fraction of the full signal, for the
aforementioned reasons.

It may appear at first glimpse that GRACE does not see sufficient altimetric signal to allow our technique to work, but that is
a trick of the eye. Fig. 4a shows the percent of altimetry's sub-monthly variability explained by the GRACE swath series.
This Percent Variance Explained (P.V.E.) is closely related to the normalization of Fig. 3c found by dividing through by Fig.
3b, and can be computed as:

$$P.V.E. = \left(1 - \frac{var(Alt - GRACE)}{var(Alt)}\right) * 100\%$$

(4)


where obviously signals other than GRACE can be inserted in its place. We see that there are large sections of ocean,
particularly the southern ocean, where the CSR swath series explains 25-75% or more of altimetry's sub-monthly variability.
It does not explain the variability within the equatorial region simply because there is so little signal there (see Fig. 3) that the
signal-to-noise ratio gets very low. The altimetric P.V.E. by ITSG2016 and ITSG2018, and by the de-aliasing models GAD5
and GAD6, are shown in direct comparison in Fig. 4. In each case, the largest P.V.E.s occur in about the same areas as in the
CSR swath series, with only the relative amplitude changing. Generally, the three high-frequency GRACE series show a
higher P.V.E. compared to altimetry than either of the GRACE de-aliasing models.

Table 1 lists the areal percentage of the ocean between 66°S and 66°N in which each series explains at least 25% of altimetry's
variance. The first data column shows this statistic for all grid cells, while the last two columns consider only those non-
coastal cells where the CSR swath series sees an RMS of at least 2 or 3 cm water height (masks outlined in Fig. 3a). All three
of the GRACE series show a large negative P.V.E. (<-25%) near 90°E, 5°N (Fig. 4a, b, c). This solid-Earth signal is the
impact of the Andaman-Sumatra earthquake from 2004, which had a large gravitational effect both during and after the event.
For statistical purposes, we have removed the most affected grids from our analysis in all of the following tables.


The double-difference plots (Fig. 5) obtained by subtracting one map in Fig. 4 from another via Eq. 3, can better show us
which data series most closely matches altimetry in the sub-monthly realm. The percentage of negative non-coastal ocean
area between 66°S and 66°N is listed in Table 2. These percentages measure the area where each alternative series explains
more of altimetry's variance than the CSR swath series does – or, in other words, places where the alternative series is more
likely than CSR swath to be correct. The statistics are again given for all grid cells as well as for the low-signal and high-



signal areas separately. They again omit the earthquake area as well as coastal regions which might contain ice or hydrological leakage effects.

We use the CSR swath series as our main comparison series here. We immediately note that the CSR swath series is, in nearly all places, better than the AOD5 model upon which it was founded. Fig. 5a is clear proof that the CSR swath series is not merely regurgitating the apriori ocean model provided to it, but is altering it in a manner which makes it more like altimetry – a manner very likely to be an improvement.

By comparing Fig. 5b to 5a, we see that in most places, the newer AOD6 model improves upon AOD5, making the differences

between it and the GRACE CSR swath series smaller. However, this is surprisingly not true within one region, near 210°E, 60°S. For an unknown reason, in the bright-pink-colored region there, the AOD6 model is found to be both different from the CSR swath series and measurably worse based on the altimetric P.V.E. To confirm that this difference was not caused by an error in the altimetry product, we also ran the same analysis on the high-frequency ocean de-aliasing model used by Jason altimetry (Fig. 5c). We found that the CSR swath series is roughly equivalent to the Jason ocean de-aliasing model in that

spot. This implies that GRACE, AOD5, and the Jason de-aliasing model all see one signal, which additionally matches with the altimetry data, while the AOD6 model sees something very different. Presumably the AOD6 model is wrong, though precisely why is an ongoing question.

We examined whether an error over a limited time period was causing the AOD6 discrepancy, but found that the same patch

of poorly-matched data occurred for all years from 2003-2016. Fig. 6a shows the regionally-averaged sub-monthly time series of AOD5, AOD6, and CSR swath over a single year, compared to the sea level anomalies from Jason altimetry. The time-series of all four series are very similar, with altimetry being the most unlike the other three, as might be expected. The correlation between CSR swath and AOD5 time-series is 94.2%, while the correlation between CSR swath and AOD6 is a still-high 91.6%. However, the main difference is not a matter of correlation, but of amplitude: AOD6 is magnified compared

to AOD5. The standard deviation of the CSR swath series over 2003-2013 is 3.72 cm, which corresponds well with AOD5's standard deviation of 3.75 cm. The AOD6 time-series has a much higher standard deviation than either: 5.33 cm.

We then ran a spectral analysis of the four time series over the mostly-continuous 2003-2013 timespan (Fig. 6b), using the least-squares-based Lomb-Scargle method to accommodate the remaining gaps in the data. From this we learned that the

AOD6 differences are not caused by a change at a single harmonic, but are rather an amplified signal throughout the entire sub-monthly band, particularly at periods below 10-15 days. The root cause of this difference is currently unknown to us, though the comparison with altimetry strongly suggests that the AOD6 data is in error in this region.





We also compared the CSR swath series to the two ITSG high-frequency GRACE series (Fig. 4d and 4e). The older ITSG2016
series is generally less like altimetry than the CSR swath series, particularly in the equatorial and northern oceans. Only 26.8%
of the ocean area shows an improvement relative to altimetry, when switching from CSR swath to ITSG2016. The newer
ITSG2018 series is decidedly better, with 45.2% of the grids improving over CSR swath, including 73.7% of the area with
more than 2 cm RMS GRACE variability. ITSG2018 is likely an improvement over the CSR swath series in the southern
ocean, while CSR swath is probably still better in the equatorial and quieter northern parts of the ocean. Larger differences
for both ITSG series along some coasts suggest either worse tidal models or (particularly near Greenland) significantly
increased leakage from nearby land, in comparison to the CSR swath series. We would also anticipate the arctic and near-
Antarctic oceans to be more poorly measured by ITSG2018 than CSR swath because of the large impact of ice leakage in those
areas, but that cannot be tested using the non-polar Jason altimetry data.


## 5 Band-passed Results

We used a set of Gaussian temporal windows to act as band-pass filters, dividing the above results into three pieces: signals
with periods below 10 days, between 10-20 days, and between 20-30 days. To provide a baseline, Fig. 7 shows the altimetric
P.V.E. by the CSR swath series for each frequency band. For signals with periods of more than 10 days, the CSR swath series
perceives at least 25% and often more than 50% of the altimetric variability across most of the southern ocean and the high-
signal part of the north Pacific. Conversely, the swath series does a poor job of reproducing the altimetry signal at periods
shorter than 10 days. Since most of the altimetric signal in this highest temporal frequency band occurs over small spatial
scales along the major currents, it is not surprising that GRACE cannot see it.


Fig. 8 shows the double-difference P.V.E. comparison for the three frequency bands, with statistics given in Table 3. Its left-
hand column depicts the CSR swath series minus AOD5 results, demonstrating that the CSR swath series estimates an
improved ocean signal across the entire southern ocean in all three frequency bands, but especially for signals with periods
longer than 10 days. In the 10-20 day band, the CSR swath series better explains the altimetric signal than its apriori model
can in 83.4% of the high-signal (RMS > 2 cm) part of the southern ocean. In the 20-30 day band, that improves to 87.8%.

The middle column of Fig. 8 shows the same thing, but for AOD6. For periods shorter than 10 days, the two series are roughly
equivalent. For sub-monthly periods longer than 10 days, the AOD6 series better matches altimetry over northern and
equatorial regions, while the CSR swath series proves to be 10% or more better than AOD6 in much of the southern ocean.
Improvements over its AOD5 predecessor can be seen in all frequency bands. The CSR swath series better explains the 10-





20-day altimetric signal than the AOD6 model does in 44.5% of the high-signal southern ocean bins. For the 20-30 day band, CSR swath better explains 58.9% of these bins. Additionally, the small region of poor performance near 210°E, 60°S is confirmed to be a very high frequency issue, mainly visible in the 1-10-day band and not visible in the 20-30-day band.

330 The right-hand column of Fig. 8 is the ITSG2018 comparison. (ITSG2016 was found to be inferior to its successor in all bands, so we do not depict it here.) There are few frequency-based distinctions between the CSR swath and ITSG2018 series. In the southern ocean, ITSG2018 signals with any sub-monthly period are more likely than CSR swath signals to be like altimetry (ITSG2018 is better in 65.5%, 85.6%, and 79.4% of the grids by area, for the 0-10, 10-20, and 20-30 day bands respectively). The opposite is true in the lower-signal equatorial and northern oceans (similar percentages of 26.0%, 45.2%,

335 and 44.7%). The large positive (>30%) coastal differences near Alaska, Patagonia, the Antarctic Peninsula, and Greenland again suggest that ITSG2018 does not segregate the land ice leakage as well as the CSR swath series does, while the large negative (<-30%) coastal differences north of Australia hint towards a possibly improved tidal or ocean model. CSR swath series' generally higher equatorial P.V.E. could indicate that series has better reduced the GRACE stripe-like errors compared to ITSG2018 in these regions of relatively low oceanographic signal.


## 6 Conclusions

Using a comparison with Jason altimetry, we have demonstrated that two modern near-daily GRACE series are capable of seeing real sub-monthly oceanographic signal. Both the CSR swath series and ITSG2018 explain a fair proportion of the high-

345 frequency altimetric signal outside of major currents, more of it than the apriori ocean models they are based upon can. We found that these near-daily GRACE series are particularly sensitive for signals with periods above about 10 days, and less so as the signal length shortens below that.

At the moment, it appears that the CSR swath series is better able to eliminate land-leakage from ice and hydrology from entering the ocean grids, making it distinctly better than ITSG2018 along the coasts and near the large ice sheets. CSR swath

350 also appears probably better at removing the false north-south stripes commonly created during GRACE processing, allowing the quieter equatorial regions to be better estimated. The ITSG2018 series, on the other hand, appears to better observe the sub-monthly state of the southern ocean, particularly in those areas with large amounts of variability. This might be an impact of the different processing scheme, or it might be due to the use of the newer AOD6 model as an ocean de-aliaser. To test the

355 latter, we plan on eventually reproducing the CSR swath data with AOD6 as an apriori model.





Our analysis here demonstrates that, particularly in the poorly-observed southern ocean, sub-monthly GRACE data can be used to improve our knowledge of the ocean bottom pressure. We hope to use this in the future to validate global ocean models, perhaps even merging the GRACE data with modeled results in order to produce a combined de-aliasing product superior to either source alone.

# 7 Data Availability

All data required to reproduce this work is freely available online. The Jason-1 GDR-E records from 2002-2008 are available at ftp://podaac.jpl.nasa.gov/allData/jason1/L2/gdr_netcdf_e/. The Jason-2 GDR-D records from 2009-2016 are available at ftp://ftp.nodc.noaa.gov/pub/data.nodc/jason2/gdr/gdr/. The GRACE CSR Swath data is not generally available for release yet, but a subset consisting of only the ocean grid cells between 66°S and 66°N has been placed at <A website will be provided here before publication>. The GRACE ITSG2016 series can be found at https://www.tugraz.at/institutes/ifg/downloads/gravity-field-models/itsg-grace2016/ and the ITSG2018 is here: https://www.tugraz.at/institutes/ifg/downloads/gravity-field-models/itsg-grace2018/. The GRACE de-aliasing AOD5 and AOD6 model data can be downloaded from ftp://rz-vm152.gfz-potsdam.de/grace/Level-1B/GFZ/AOD/. The Jason de-aliasing DAC model data can be found either as the "HF" section of the GDR files mentioned above, or else in combination with the IB effect at https://www.aviso.altimetry.fr/en/data/products/auxiliary-products/atmospheric-corrections/description-atmospheric-corrections.html.

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





**Figures**

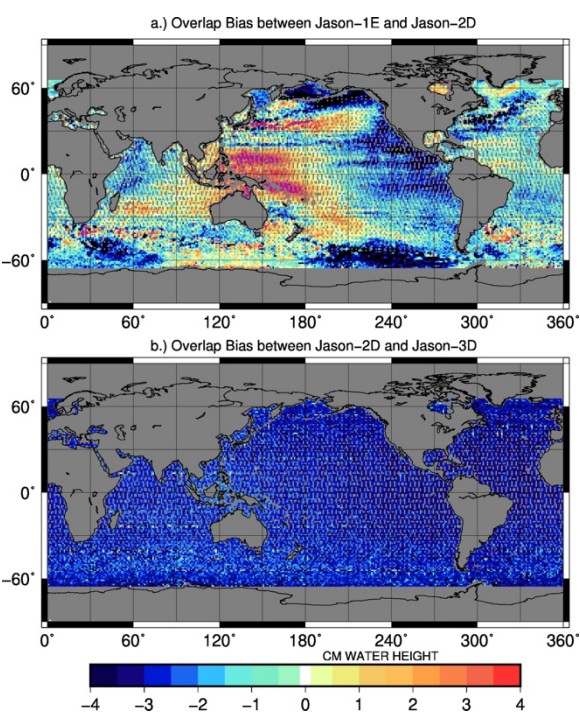

**Figure 1: Bias offsets between (a) Jason-1 and Jason-2, and (b) Jason-2 and Jason-3.**



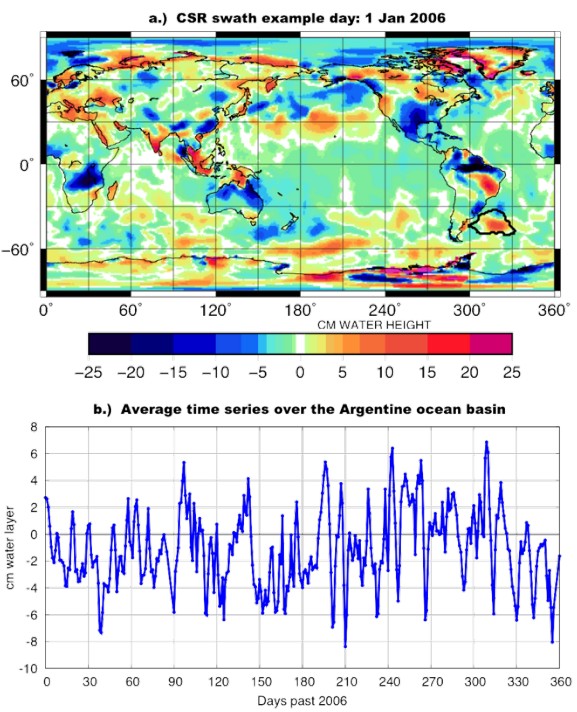


**Figure 2:** **(a) An example day (1 Jan 2006) of the GRACE swath data. (b) An example of the temporal resolution of the GRACE swath series, over the Argentine ocean basin, off the coast of Brazil. The thick line in (a) shows the basin outline used.**





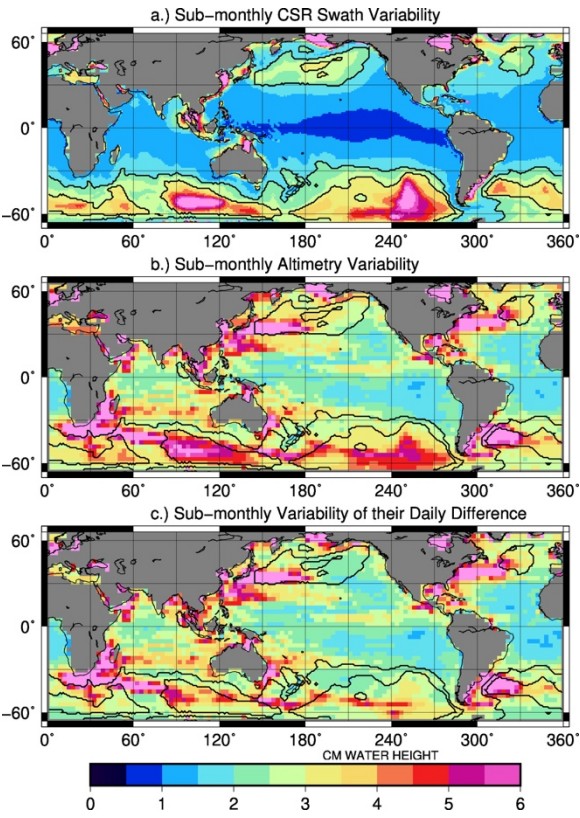

**Figure 3:** **Standard deviation of (a) CSR Swath sub-monthly ocean bottom pressure, (b) Jason altimetry sub-monthly sea surface height anomalies, and (c) the difference between the two series. Units are cm of water height in all cases. For comparison's sake, all plots include the 2cm and 3cm contour lines computed from the CSR swath plot (a).**





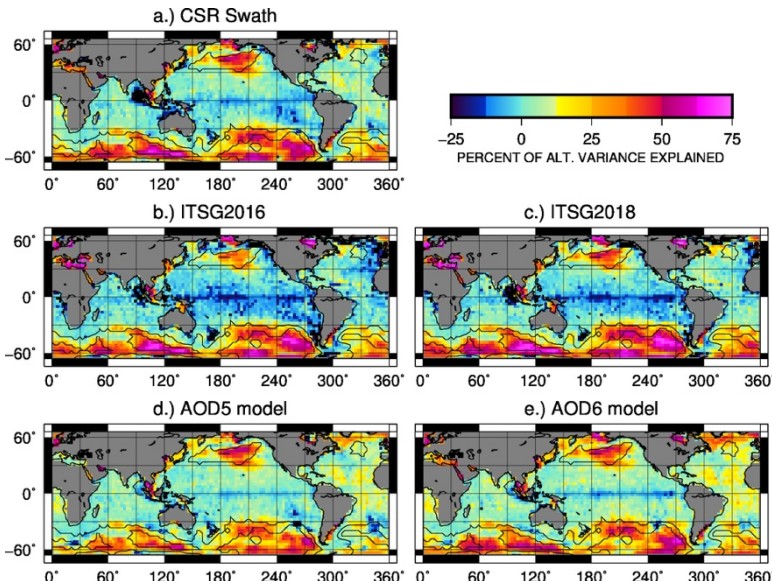

**Figure 4: Percent of altimetric variance explained by five different series: (a) CSR swath GRACE data, (b) ITSG2016 GRACE data, (c) ITSG2018 GRACE data, (d) AOD5 ocean de-aliasing model, and (e) AOD6 ocean de-aliasing model.**

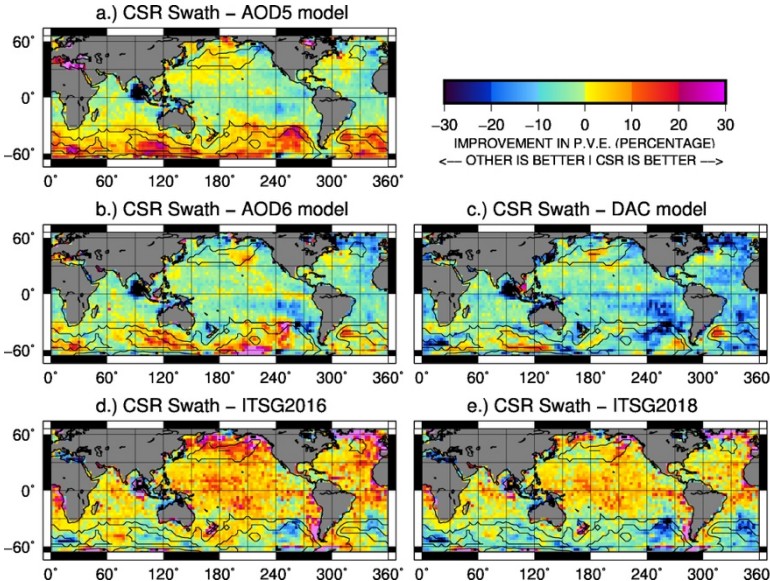

**Figure 5: Differences in sub-monthly P.V.E for CSR swath vs. another data type: (a) AOD5 de-aliasing ocean model, (b) AOD6 de-aliasing ocean model, (c) altimetry's DAC de-aliasing ocean model, (d) ITSG2016 GRACE data, and (e) ITSG2018 GRACE data. Values are relative to CSR swath P.V.E.s, so positive numbers (red) denote that CSR swath matchest altimetry better, while negative numbers (blue) denote that the other series matches altimetry better.**





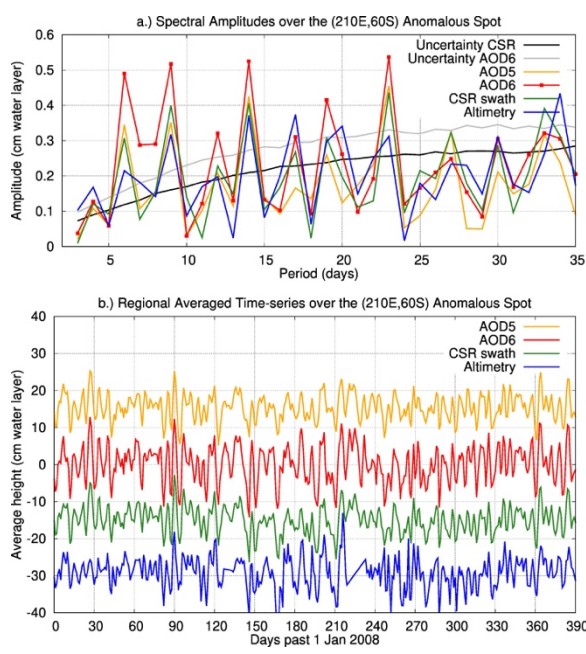

**Figure 6:** **Example timeseries (a) and spectral amplitude plot over AOD6's anomalous spot near (210°E, 60°S), with lines offset for clarity. Spectral analysis (b) was computed for the timespan 2003-2013. The black line shows the uncertainty for the CSR swath series (which is similar to the AOD5 and altimetry uncertainties), and the grey line shows the uncertainty for the AOD6 series. Signals below these lines are not statistically significant.**






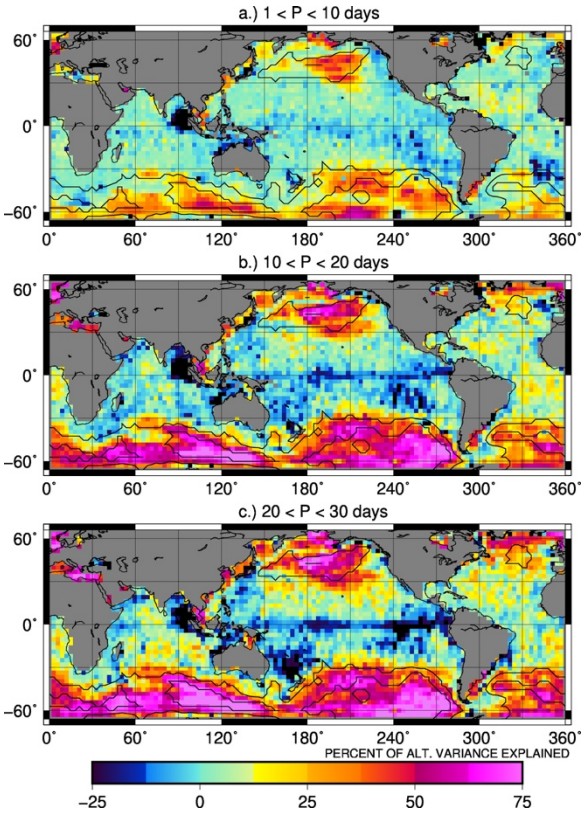

**Figure 7:  Percent of altimetric variance explained by the CSR swath series at three sub-monthly frequency bands: (a) periods below 10 days, (b) periods between 10-20 days, (c) periods between 20-30 days.**



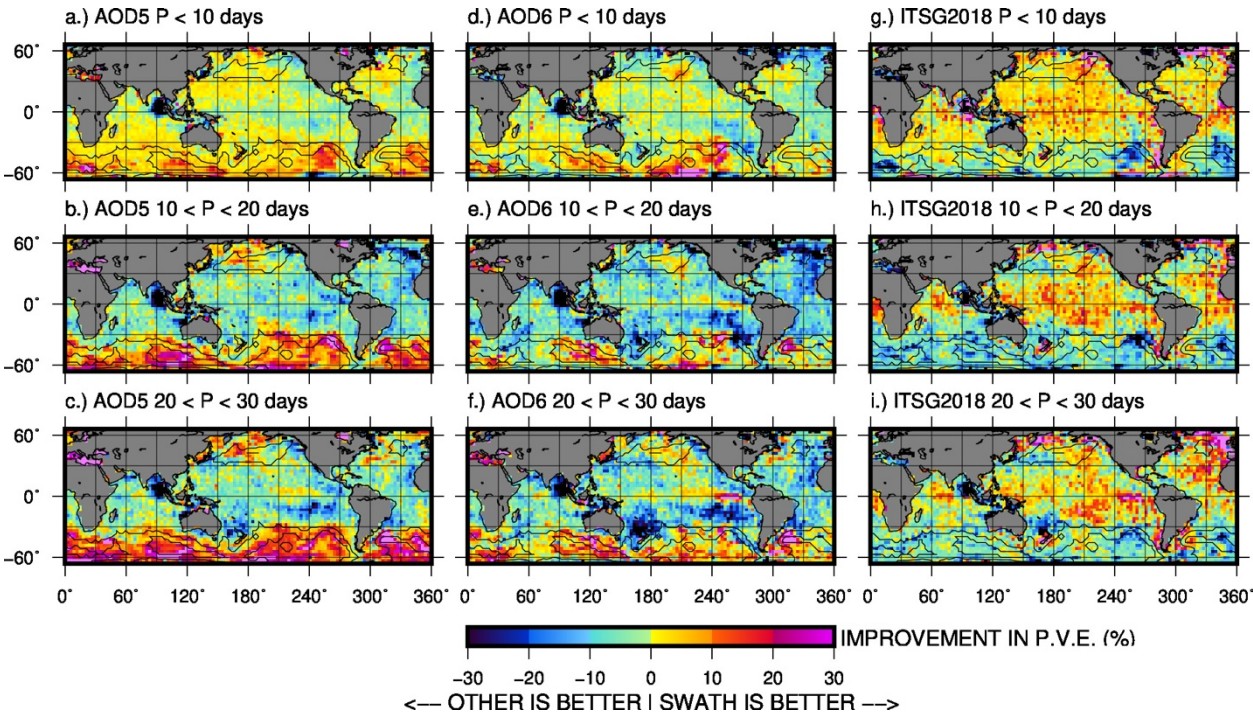

**Figure 8: Differences in P.V.E for CSR swath vs. another data type, per frequency band: (a, b, c) AOD5 de-aliasing ocean model, (d, e, f) AOD6 de-aliasing ocean model, and (g, h, i) ITSG2018 GRACE data. The top row gives the highest sub-monthly frequency band, while the bottom gives the lowest sub-monthly frequency band. Values are relative to CSR swath P.V.E.s, so positive numbers (red) denote that CSR swath matchest altimetry better, while negative numbers (blue) denote that the other series matches altimetry better.**






## Tables

**Table 1: Percent of non-coastal ocean area explaining at least 25% of altimetry's sub-monthly variance. 2 cm and 3 cm RMS bounds are defined in Figure 3a.**

| Data series | All grid cells | RMS > 2 cm | RMS > 3 cm |
|---|---|---|---|
| CSR swath | 17.1% | 43.1% | 66.2% |
| ITSG2016 | 16.2% | 40.6% | 68.7% |
| ITSG2018 | 19.4% | 48.9% | 77.2% |
| AOD5 | 13.0% | 31.8% | 50.1% |
| AOD6 | 17.3% | 41.3% | 58.1% |

**Table 2: Percent of non-coastal ocean area where the altimetric variance is better explained by the given time-series than by the CSR swath series. 2 cm and 3 cm RMS bounds are defined in Figure 3a.**

| Data series | All grid cells | RMS < 2 cm | RMS > 2 cm | RMS > 3 cm |
|---|---|---|---|---|
| ITSG2016 | 26.8% | 20.9% | 39.1% | 56.7% |
| ITSG2018 | 45.2% | 31.6% | 73.7% | 85.3% |
| AOD5 | 52.2% | 71.1% | 12.4% | 4.1% |
| AOD6 | 70.1% | 79.9% | 49.4% | 33.3% |
| DAC | 87.5% | 91.8% | 78.5% | 70.3% |



**Table 3: Percent of non-coastal ocean area where the altimetric variance is better explained by the given time-series than by the CSR swath series. 2 cm RMS bounds are defined in Figure 3a. Column 5 estimates only over the part of those regions which are in the southern ocean.**


| Data series | Frequency Band | All grid cells | RMS > 2 cm | RMS > 2 cm southern |
|---|---|---|---|---|
| AOD5 | 1-10 days | 39.0% | 12.9% | 8.6% |
|  | 10-20 days | 65.1% | 25.9% | 16.6% |
|  | 20-30 days | 58.5% | 20.2% | 12.2% |
| AOD6 | 1-10 days | 58.7% | 43.8% | 40.0% |
|  | 10-20 days | 79.4% | 60.0% | 55.5% |
|  | 20-30 days | 68.6% | 45.7% | 41.1% |
| ITSG2018 | 1-10 days | 38.0% | 61.0% | 70.5% |
|  | 10-20 days | 57.4% | 82.1% | 87.8% |
|  | 20-30 days | 54.0% | 73.0% | 79.5% |