# Peer review of "Evaluation of Sub-Monthly Oceanographic Signal in GRACE "Daily" Swath Series Using Altimetry"

_Ocean Science, 2019_

## Referee Comment (RC1) · Henryk Dobslaw (Referee) · 16 Jul 2019

The presented manuscript discusses bottom pressure variations at periods between 2 and 30 days picked up by satellite altimetry and gravimetry mission in comparison to corresponding predictions from numerical ocean models. The paper demonstrates that time-series of large-scale gravity models provide information on sub-monthly variability not introduced into the processing from a priori de-aliasing models. Moreover, the authors identify and characterize a potential artifact in the AOD1B RL06 background model data. The study generally fits into the scope of the journal, is well designed, and leads to convincing results. I nevertheless see a number of points that might be

improved before publication.

It is stated at numerous places in the paper that the ITSG series are governed by external geophysical models whereas the swath solutions are free of a priori model information. I believe that both claims are not entirely correct and should be relaxed in the sense that both solutions utilize some external "information" (in certainly distinctly different ways). Note that such information might also include the assumptions that variability over oceans and land is uncorrelated, and that ocean bottom pressure variability in the tropics is very small. The inverse problem with just a 24-hour subset of GRACE data is ill-posed and needs to be stabilized in some way to obtain a reliable solution.

It is nice to see that swath solutions show less noise in the tropics than ITSG, but it should be acknowledged at some point in the paper that reducing noise in regions where geophysical signals are expected to be non-existant can be very easily achieved by regularizing the solution towards zero. In case of an unexpected event at some later date (say, an earthquake), regularized solutions tend to underestimate or even miss that signal. Maybe the authors could elaborate a little further about the utilization of regularization (or related techniques) in the swath solutions when discussing the tropical oceans for the pleasure of their geodetic audience?

The Low-degree Stokes coefficients not accessible from GRACE alone can be assumed to vary rather slowly in time so that linear interpolation from monthly to daily sampling might be feasible. Have you tried this in some way? Would you expect any consequences for your conclusions? Which regions might be affected most?

The impact of change in the MSS model might be explored a little further. What is the difference between the 16y and 20y MSS? Is that effect perfectly linear, or do you see larger biases in regions where the MSS models differ most?

The discussion of the signal in the Zapiola Gyre is interesting and deserves more attention. There has been previous work about the dynamics seen from both altimetry

and gravimetry (see 10.1029/2018JC014189 and references therein), and it would fit well into the scope of the journal if some further discussion is added based on the swath data.

The assessment of the anomaly present in AOD1B RL06 in the South Pacific appears to be sound and forms valuable feedback for the development of general ocean circulation models. Our present-day understanding is that an overly simplyfied Ross Sea bathymetry in the MPIOM model run (i.e., all ocean areas covered by shelf-ice are treated as land) distorts the dominant eigenmodes in the larger region at periods around 3 to 8 days. I expect to see this problem reduced to a large extent in the next release of AOD1B.

A number of minor points might be also considered during the revision:

Section 4: In terms of the language, I suggest to clearly separate between observations, which might "see" or "observe" signals; and on the other hand numerical models, which rather "predict" variations. I suggest to modify this throughout the whole manuscript, but in particular adapt the wording in Section 4.

l. 135: The products used here have higher resolution in time but not in space, right?

l. 137: The desire of labelling the CSR swath data as the "main" GRACE product is understandable, but not fully justified. Maybe just call it "your" GRACE product?

l. 159: ... results shown here represent the full non-tidal mass signal.

l. 174: Who, in fact, is Norbert?

---

## Author Comment (AC1) · 17 Sep 2019

Thank you, Henryk, for your very helpful review. We've altered the text in several places because of it. The attached document contains our more detailed replies. Please let us know if there's anything else you'd suggest changed.

We have tentatively added a line in the manuscript which states:

"This assessment has recently been provided to the creators of the AOD6 model, who hypothesize that the problem may be caused by an error in the MPIOM ocean model run near the Ross Sea. They now believe that their model was treating the ice shelf as

land, which distorted the region's dominant eigenmodes at 3-8 day periods. They are working to correct this flaw for the next release of the de-aliasing product."

Please let us know if you and your colleagues would appreciate having such an explanation given, or not. We will abide by your wishes on that.

– Jenni and Himanshu

Please also note the supplement to this comment:
https://www.ocean-sci-discuss.net/os-2019-68/os-2019-68-AC1-supplement.pdf
* * *
[Figure]

**Supplement:**

Thank you, Henryk, for your very helpful review. We've altered the text in several places because of it. Here are our more detailed replies. Please let us know if there's anything else you'd suggest changed.

-- Jenni and Himanshu

1.) It is stated at numerous places in the paper that the ITSG series are governed by external geophysical models whereas the swath solutions are free of a priori model information. I believe that both claims are not entirely correct and should be relaxed in the sense that both solutions utilize some external "information" (in certainly distinctly different ways). Note that such information might also include the assumptions that variability over oceans and land is uncorrelated, and that ocean bottom pressure variability in the tropics is very small. The inverse problem with just a 24-hour subset of GRACE data is ill-posed and needs to be stabilized in some way to obtain a reliable solution.

We have tried to be more careful in our wording. You are correct that some sort of stabilization is obviously required, and we now mention that in the paper. What we intended to say with regards to the CSR swath solution, was that there is no external geophysical model used to inform any constraints. The constraints are purely driven by GRACE information. The only external information used to define the constraints is the land and ocean boundary. Our information concerning your understandable question about "the variability over the oceans in the tropics is very small" is also derived purely from GRACE. While the inverse problem with just 24 hr subset of the GRACE data is ill-posed, the stabilization is helped by the fact that only the mascons under the ground tracks are estimated for the day. This inverse problem not as ill-posed as a global inversion from a 24 hr data. We have attempted to make this more clear in the paper:

"The global mascon solutions and regularization are purely driven by GRACE without any influence from external models. The only external information used to inform the constraints is the land/ocean boundary mask. All the other information for constraints comes from expected signals in GRACE for that month from regularized spherical harmonic solutions (Save et. al. 2016) and the GRACE groundtrack. Since the daily constraints are derived from the respective monthly expected signals from GRACE, the regularization also allows for adjustment of unexpected signals that are captured the monthly solutions. The only submonthly signals that will get constrained to zero in the swath solutions are the signals that may have a zero mean over 30 days throughout the mission but do have sub-monthly variability. The implementation of the swath estimation assumes that such locations are very rare. Thus, the time-variable regularization process used does not bias or attenuate future regional signals based on statistics from models or past GRACE months, but is intentionally designed to encourage no land/ocean correlation in order to reduce leakage. Further details of the data processing for producing the daily GRACE swath solutions is available in Save et. al. 2018 (in-preparation/in-review)."

There isn't a lot of detailed information about how the ITSG series are regularized, but everything we've read/heard suggests that they do it based on signals (RMS, etc) from apriori models.  We sent an email to Torsten/Andreas asking for information, and this is what Andreas replied:

The big picture of the daily processing has not changed much since Enrico's paper in 2012 (https://doi.org/10.1016/j.jog.2012.02.006). We still use daily GRACE normals (in spherical harmonics) and constrain them using a stochastic model derived from geophysical model output, so most of the conclusions from back then can still be applied today. What became more sophisticated over the years is the way how the constraints are computed. We put a lot of thought into how the covariance function of a high-dimensional stationary process can be robustly estimated. This mainly involves exploiting geophysical properties, for example, land/ocean masks.

We've updated the text to make this more clear, as well as adding the reference into it for those who want more information.

2.) It is nice to see that swath solutions show less noise in the tropics than ITSG, but it should be acknowledged at some point in the paper that reducing noise in regions where geophysical signals are expected to be non-existant can be very easily achieved by regularizing the solution towards zero. In case of an unexpected event at some later date (say, an earthquake), regularized solutions tend to underestimate or even miss that signal. Maybe the authors could elaborate a little further about the utilization of regularization (or related techniques) in the swath solutions when discussing the tropical oceans for the pleasure of their geodetic audience?

There is a main swath paper ready to be submitted for review shortly that will discuss all the details of the constraints, etc.  We have now included a few details about the regularization in this paper, as well as adding a citation toward the in-progress work for further information in the future.  The regularization matrix for the swath solutions are essentially an extension of the monthly regularization matrix design process (as described in Save et. al. 2016) but also includes information for the ground track.

As for your concerns about the tropics, you are correct that it's easy to "regulate away" errors by just driving them the full signal+error to zero.  We see no indications that this is what's happening in the swath solutions, though, as said in above.  The only way signal (or noise) in the tropics could be artificially driven to zero is if the monthly mean signal was zero, but the sub-monthly non-noise signal wasn't.  That's unlikely to be commonplace, so if the tropics show low sub-monthly noise, it's because there's also low monthly-scale signal there within that particular month.

3.) The Low-degree Stokes coefficients not accessible from GRACE alone can be assumed

to vary rather slowly in time so that linear interpolation from monthly to daily sampling might be feasible. Have you tried this in some way? Would you expect any consequences for your conclusions? Which regions might be affected most?

If you're talking about the geocenter and J2 terms, we toyed with interpolating them, by fitting a trend/annual and interpolating based on that model. We chose not to add that complication in this paper, however, since we're looking at the sub-monthly signal. As you say, low degrees like the geocenter aren't likely to change rapidly, even assuming we had good daily geocenter data to represent reality with. And if we simply interpolated the change linearly between months, the change in the sub-monthly frequency band would be zero. Thus the omission. We've added a comment about this in section 3.2.

4.) The impact of change in the MSS model might be explored a little further. What is the difference between the 16y and 20y MSS? Is that effect perfectly linear, or do you see larger biases in regions where the MSS models differ most?

At some point in the future, I (Jenni) really would like to dig up my own MSS model and reprocess all my altimetry series correctly. But I haven't done that yet and it's no doubt going to be a pain. I've tried to look into the two MSS models used by Jason, but getting information about them has also proved unexpectedly difficult. The sum total of all the info I can find is from the AVISO website here:
https://www.aviso.altimetry.fr/en/data/products/auxiliary-products/mss.html
As you'll note, it's not extensive, nor does it link to any more useful papers. So I don't actually know the answer to your first question.

But I'm less concerned about short-term, mostly-regional differences where the two different models see slightly different signals, than I am about the bias difference. As best Don and I can figure, the main reason our Figure 1a sees such a huge signal isn't because the models are necessarily very different, but simply because they're centered at a different time. I'm attaching a little sketch I drew to show you what I mean. Even assuming the two MSS models were perfectly identical, because of the 16-year vs. 20-year time span, they're going to see different MEAN values. That's where the bias jump between missions (or rather, between MSS models) is coming from. It's way too big to simply be from real model improvements. They just didn't recenter the bias to the same timespan as the old MSS model. And also didn't tell anyone, which is even more frustrating – and why I wanted to explicitly mention it in this paper, so at least the information is out there someplace.

[Figure]

(I rather doubt that the journal would appreciate publishing my hand-drawn picture, alas.)

5.) The discussion of the signal in the Zapiola Gyre is interesting and deserves more attention. There has been previous work about the dynamics seen from both altimetry and gravimetry (see 10.1029/2018JC014189 and references therein), and it would fit well into the scope of the journal if some further discussion is added based on the swath data.

We absolutely agree that this is a fascinating area.  There are some interesting results from the Gyre being included in the main swath paper.  The summary is that GRACE swath solutions can clearly observe rotation in the gyre at a sub-monthly frequency that has been previously seen in the altimeter data.  We're hoping to do more work in this region in the future – possibly with both ITSG2018 and CSR swath, since they both seem to give plausible localized results.  That's a bit outside the scope of this paper, however – and probably shouldn't be published until after Himanshu's swath paper, anyhow.

Also, thank you for the link.  That was an excellent paper.

6.) The assessment of the anomaly present in AOD1B RL06 in the South Pacific appears to be sound and forms valuable feedback for the development of general ocean circulation models. Our present-day understanding is that an overly simplyfied Ross Sea bathymetry in the MPIOM model run (i.e., all ocean areas covered by shelf-ice are treated as land) distorts the dominant eigenmodes in the larger region at periods around 3 to 8 days. I expect to see this problem reduced to a large extent in the next release of AOD1B.

Congrats on figuring out the bug!  I'm glad to hear it.  What a bizarrely localized issue.

Is this just for our own information, or would you like us to quote you in here so everyone knows?  We assume the latter and have altered the manuscript accordingly, but if you'd prefer for us not to do so, that's also fine.  Let us know what you'd like and we'll see it done.

Section 4: In terms of the language, I suggest to clearly separate between observations, which might "see" or "observe" signals; and on the other hand numerical models, which rather "predict" variations. I suggest to modify this throughout the whole manuscript, but in particular adapt the wording in Section 4.

Sorry.  That's my bad.  Srinivas has only been yelling at me about this imprecision since 2002, now.  :)  You'd think I'd know better.  I've gone through and corrected it, as you suggest.

l. 135: The products used here have higher resolution in time but not in space, right?

Correct.  We altered the line to make this more clear.

l. 137: The desire of labelling the CSR swath data as the "main" GRACE product is understandable, but not fully justified. Maybe just call it "your" GRACE product?

Agreed, this is definitely not the "main" GRACE series.  We meant the main series used by THIS paper, not overall.  We have altered the text to make that clear.

l. 159: ... results shown here represent the full non-tidal mass signal.

We have added a line confirming that the ocean tide model was removed and has not been restored.

I am so sorry.  I use Mendeley to organize my citations and somehow, in the submitted version of the document, the second half of the bibliography disappeared, including all authors with last names later than "M", and the ITSG citations shifted from the first author (Mayer-Gurr) to the last (Norbert).  I have no idea how, but that has been corrected.  Sorry again.

---

## Referee Comment (RC2) · Anonymous Referee #2 · 12 Nov 2019

In oceanographic applications of GRACE, the question often arises whether the supposed gravity signal is really from GRACE measurements or is merely the prior ocean model used in the GRACE processing. The question is clearly important for a variety of ocean applications. This paper makes some useful progress in answering the question, based on three GRACE gravity solutions (although none is an "official" product from one of the U.S./German GRACE centers.

I recommend the paper be revised before it is published. Most of the items I list below are minor and easily dealt with (or dispensed with, if the authors decide). The one major item requires some data reprocessing. Another major item is: "Half the References

section is missing!" (Perhaps a problem with the OS website?).

Detailed items follow:

1. My major complaint is not with the GRACE processing, but with the altimetry – and more specifically it is with Figure 1, which reports a large "bias offset" between Jason-1 and Jason-2. This certainly will be a surprise to the altimeter community, and it contradicts what has been previously published – see, for example, papers by Ablain et al. (doi: 10.1080/01490419.2010.487805) and Beckley et al. (doi: 1.1080/01490419.2010.491029).

The Jason project teams and most users would be very concerned to see Figure 1 published as is, and for good reason.

In fact, this "bias" is merely caused by use of inconsistent versions of Jason GDRs. One cannot blindly combine different GDRs, based on different corrections and possibly other things (retracking?), and expect consistency. The authors should not rely on the "experienced aid of Don Chambers," but should carefully examine user handbooks and other documentation. They will find that there are other differences, too, not just the MSS model.

After I did some digging, I can add one thing in the authors' defense, which is a point about better data documentation. For some reason, the CLS group uses a naming (or non-naming) convention that is confusing. Their MSS evidently comes with a rate, and by changing the "reference time period" the MSS obtains different values, even though the fundamental MSS model is the "same" and retains the same name. (It is not a matter of the time span of data going into the determination of the MSS.) It would be much better if CLS didn't confuse users in this way, but that's the way matters stand. The GDR attributes give no hint of this problem, but the data handbooks do.

What the authors should have done, and should have written, is something like the following:

"We have used the best available Geophysical Data Records (GDRs) from Jason-1 and Jason-2, and applied consistent geophysical models to ensure a self-consistent time series of sea surface height anomalies across the missions. The source data are from Jason-1 version "E" and Jason-2 version "D" GDRs. Documentation for these different version numbers indicate the use of different processing standards, in particular ancillary geophysical models in the two sets of products. Most important for our investigation, we have used a consistent mean sea surface and ocean tide model. We have also used the ECMWF Reanalysis for the dry troposphere and inverse barometer corrections, as provided on the Jason-1 GDR-E, to mitigate any changes to the ECMWF operational analysis during our period of interest."

This does require some data processing. An alternative approach is not to use the GDRs at all, but instead use DUACS(Aviso) or MEASURES products, which are reprocessed data with consistent data handling since 1993.

2. The reference "Eumetsat,... (2016) for Jason-1 products isn't right, as Eumetsat had nothing to do with Jason-1.

3. Line 21: "as large as" -> "even larger than"

4. Line 23: How do you know the ocean models are poor in the Southern Ocean? If data assimilation has been used in their development, then I'd agree, but I thought OMCT and MPIOM had no assimilation. Is there another reason to think models are poor there?

5. Line 27: "predicated" is the wrong word to use here.

6. Line 163: "signal" -> "signal was"

7. Line 170: The authors here might wish to cite published work that has examined the barotropic circulation in this region. For example, work by Chris Hughes: doi:10.1029/2006JC003679

8. Line 210: Is MPIOM also forced by pressure? If not, how does this affect the

comparisons? Line 203 already notes that OMCT uses pressure forcing.

9. Line 214: The Lynch-Gray reference should be augmented (or even replaced by) Carrere et al.: doi:10.1029/2002GL016473

10. Caption to Figure 3. It would be useful to give the time intervals over which these standard deviations were computed. (In fact, I don't think I saw this in the main text anywhere either, but I may have missed it.)

11. Line 247. I would add "except for the middle and North Atlantic". It seems GRACE is not improving the prior model there.

12. Lines 280, 288: Are Figures 6a and 6b reversed?

13. Line 308: Could slightly more explanation be added here, or at least a reference? It is not obvious to me how Gaussian temporal windows are being used to form a band-pass filter.

14. Line 345: I would again suggest that it is mentioned that the Middle and North Atlantic are problem areas.

15. Section 7. Since this section already lists long URL addresses for data used, those things could be eliminated in the main text.

16. I much appreciate the color scales in (for example) Figure 5, where arrows point which way which model is superior. Very useful. Some of the figures are a bit hard to read, however, and a bit cramped. The fonts/resolution of Figure 2 seems especially fuzzy – a Word feature?

---

## Author Comment (AC2) · 20 Dec 2019

Response to Reviewer #2.

Thank you very much for your helpful review. We'll respond to each point in kind:

*0. Another major item is: "Half the References section is missing!" (Perhaps a problem with the OS website?).*

This error was noted by the first reviewer and has been corrected. We honestly have no idea how the second half of the references vanished, but it's been fixed. Oops. Sorry.

*1. My major complaint is not with the GRACE processing, but with the altimetry – and more specifically it is with Figure 1, which reports a large "bias offset" between Jason-1 and Jason-2. This certainly will be a surprise to the altimeter community, and it contradicts what has been previously published – see, for example, papers by Ablain et al. (doi: 10.1080/01490419.2010.487805) and Beckley et al. (doi: 1.1080/01490419.2010.491029).*

*The Jason project teams and most users would be very concerned to see Figure 1 published as is, and for good reason. In fact, this "bias" is merely caused by use of inconsistent versions of Jason GDRs. One cannot blindly combine different GDRs, based on different corrections and possibly other things (retracking?), and expect consistency. The authors should not rely on the "experienced aid of Don Chambers," but should carefully examine user handbooks and other documentation. They will find that there are other differences, too, not just the MSS model. After I did some digging, I can add one thing in the authors' defense, which is a point about better data documentation. For some reason, the CLS group uses a naming (or non-naming) convention that is confusing. Their MSS evidently comes with a rate, and by changing the "reference time period" the MSS obtains different values, even though the fundamental MSS model is the "same" and retains the same name. (It is not a matter of the time span of data going into the determination of the MSS.) It would be much better if CLS didn't confuse users in this way, but that's the way matters stand. The GDR attributes give no hint of this problem, but the data handbooks do.*

*What the authors should have done, and should have written, is something like the following:*

*"We have used the best available Geophysical Data Records (GDRs) from Jason-1 and Jason-2, and applied consistent geophysical models to ensure a self-consistent time series of sea surface height anomalies across the missions. The source data are from Jason-1 version "E" and Jason-2 version "D" GDRs. Documentation for these different version numbers indicate the use of different processing standards, in particular ancillary geophysical models in the two sets of products. Most important for our investigation, we have used a consistent mean sea surface and ocean tide model. We have also used the ECMWF Reanalysis for the dry troposphere and inverse barometer corrections, as provided on the Jason-1 GDR-E, to mitigate any changes to the ECMWF operational analysis during our period of interest."*

*This does require some data processing. An alternative approach is not to use the GDRs at all, but instead use DUACS(Aviso) or MEASURES products, which are reprocessed data with consistent data handling since 1993. ---*

We entirely agree as to what the problem is, and the ideal way to fix it. Unfortunately, as you say, the issue is not well-documented at all, even in the handbooks, much less on the AVISO website – which means we will not be the only people to run into this problem and be baffled by it. Anyone downloading modern GDR data will run into it, because the older versions of the data are no longer available. We cannot find a copy of the old MSS model anywhere, though the most recent MSS model data can be downloaded from the AVISO website after some contortions. Similarly, the GDR-D data for Jason1 simply isn't online anymore at either PODAAC or AVISO. GDR-E only exists for Jason2 and Jason3, not Jason1. Which makes using matching GDR versions or official MSS models effectively impossible.

We chose to make a point of this in this paper largely to make others aware of the potential problem. It's very easy to miss, because it's natural to assume that since the GDR versions out there are the only ones available, they can be strung together safely. People have read the papers you suggested, and are thus

using codes which assume the mission-to-mission offset is spatially-uniform (as it otherwise would be). But with the jump between GDR-D and GDR-E, it's not.

The big problem is not that the SSH model changed between versions (that's expected), but that they didn't level it so that the (arbitrary) time-means were the same between versions. I hand-drew a cartoon of the issue below, in case that better helps explain what I'm describing. Even assuming the two MSS models were perfectly identical, because of the 16-year vs. 20-year time span, they're going to see different MEAN values. That's where the bias jump between missions (or rather, between MSS models) is coming from. It's way too big to simply be from real model improvements. They just didn't recenter the bias to the same timespan as the old MSS model, or provide a way for others to do so afterwards.

[Figure]

The resulting bias will result in incorrect results on the order of +/- 4cm heights in some areas, so it's important that others know that it needs to be handled. Ideally, of course, the producers of the GDRs would update all the Jason satellites together, or use MSS models with identical global means, to avoid this issue altogether. But they haven't, so GDR users have to deal with the issue on their own. Which they certainly can't do, if they aren't told there's an issue to begin with!

Now, as you say, the best method to handle this would be to reprocess with my own, consistent MSS model, not the differing versions inside the GDR files. Most likely we'll end up doing that for future work, if the Jason2 and 3 GDR products don't come out with a version E soon (as we keep hoping they will!). But that's going to take a lot of time and effort to code up from scratch and run, and the mean-bias correction we've already made corrects for the worst of the problem – the bias jump between missions – already. (The E-version MSS may also be more accurate on a point-by-point basis, but that just means a possible quality degradation between missions – and we already have a degradation over time with GRACE anyhow, so that's tolerable.)

As you say, DUACS is another option, but we would prefer to avoid using a premade gridded product. While that would fix this MSS issue, we have no idea how the optimal interpolation used in the combination of multiple satellites will alter the high-frequency data, particularly in areas with less good coverage. That seems to us a harder to handle problem than the change in MSS model (once the jump between missions is handled).

Because of the confusion both reviewers showed for this subject, we have lengthened and clarified this section in the text. Hopefully the details will make more sense now.

*2. The reference "Eumetsat,... (2016) for Jason-1 products isn't right, as Eumetsat had nothing to do with Jason-1.*
You're right. But actually, when altering this section of text, we realized that the Jason-1 handbook has not been correctly updated to list the current MSS information, so we instead pointed to the AVISO website, which we presume has the most up-to-date information.

*3. Line 21: "as large as" -> "even larger than"*

Corrected

*4. Line 23: How do you know the ocean models are poor in the Southern Ocean? If data assimilation has been used in their development, then I'd agree, but I thought OMCT and MPIOM had no assimilation. Is there another reason to think models are poor there?*

It's true that neither OMCT nor MPIOM use data assimilation, but at the same time, their creators do test them (or maybe even "tune" them?) by checking against whatever data exists. So in areas where data is limited, it's hard for them to determine if the models go amiss in some way, and correct for it. Thanks to its depth and difficulty to get to, the southern ocean is one of those places which is not well observed (few bottom pressure recorders, limited XBT drops, etc). And thanks to its uneven topography and global zonal circulation, its physics is complex. All of which make errors more likely than elsewhere.

The easiest way for me to demonstrate the heightened uncertainty in this area is simply to show you the standard deviation of the submonthly differences between AOD1B RL05 (OMCT) and AOD1B RL06 (MPIOM):

[Figure]

Notice that some of the biggest submonthly differences – up to 5cm in equivalent water height – are in the Southern Ocean. The larger uncertainties in this area are confirmed by the AOD1B modelers (ie: Dobslaw et al 2017; doi:10.1093/gji/ggx302) who similarly see large differences in their models and improved GRACE KBRR residuals in the area. Other modelers recognize the area as similarly less well-known.

The usual assumption is that a newer model is generally better – and in fact, we've proved that in this paper, in terms of AOD1B. But, as we also showed here, there are remaining submonthly errors in the model. It's hard to say exactly how large those errors are (I couldn't find any good paper on it for any of the models used here), due to the paucity of observed data in the region, but it's telling that neither the OMCT nor MPIOM modelers were surprised to hear about potential weaknesses in their models in this area.

*5. Line 27: "predicated" is the wrong word to use here.*

Corrected

*6. Line 163: "signal" -> "signal was"*

Corrected

*7. Line 170: The authors here might wish to cite published work that has examined the barotropic circulation in this region. For example, work by Chris Hughes: doi:10.1029/2006JC003679*

Yes, thank you, done. That was, in fact, the paper that got me (Jennifer) interested in this region.

*8. Line 210: Is MPIOM also forced by pressure? If not, how does this affect the C3 comparisons? Line 203 already notes that OMCT uses pressure forcing.*

Yes, both models use comparable types of forcing, including pressure. We've altered the text to make this more explicit.

*9. Line 214: The Lynch-Gray reference should be augmented (or even replaced by) Carrere et al.: doi:10.1029/2002GL016473*

Done, thanks.

*10. Caption to Figure 3. It would be useful to give the time intervals over which these standard deviations were computed. (In fact, I don't think I saw this in the main text anywhere either, but I may have missed it.)*

Oops. You're right. That's now fixed, both in Fig 3 and in the main text. The span used was April 2002 to January 2016 (the periods when we had data for all the series). Sorry.

*11. Line 247. I would add "except for the middle and North Atlantic". It seems GRACE is not improving the prior model there.*

Agreed, corrected. It's interesting that there's such a neutral response in the Atlantic, actually.

*12. Lines 280, 288: Are Figures 6a and 6b reversed?*

Yes, thanks. Corrected.

*13. Line 308: Could slightly more explanation be added here, or at least a reference? It is not obvious to me how Gaussian temporal windows are being used to form a band-pass filter.*

Done. Basically, you can use two windows of different lengths to create a band-pass filter. For example, if you process the same series with first a 30-day boxcar sliding window and (separately) a 20-day boxcar, then you can difference the 20-day series from the 30-day one to get a bandpass filter between 20-30 days. It's the old game where windowing in the time domain is equivalent to filtering in the frequency domain. We're using Gaussian windows to avoid ringing due to sharp cutoffs, but the same principle applies. (Yes, we've checked this with an FFT in the past.) The benefit of using windows rather than frequency-based techniques is that it allows you to work with gappy data.

*14. Line 345: I would again suggest that it is mentioned that the Middle and North Atlantic are problem areas.*

Done.

*15. Section 7. Since this section already lists long URL addresses for data used, those things could be eliminated in the main text.*

Good idea. Done.

*16. I much appreciate the color scales in (for example) Figure 5, where arrows point which way which model is superior. Very useful. Some of the figures are a bit hard to read, however, and a bit cramped. The fonts/resolution of Figure 2 seems especially fuzzy – a Word feature?*

Yes, it annoys us, too, and yes, Word makes it worse than the file is on its own.  Figure 2 is a combination of a line plot (made in gnuplot) and a map (made in GMT).  We can't find a "pretty" way of joining the two, except doing so manually in a graphics program, which reduces the resolution – and forces us to save in PNG format (since the journal won't accept jpgs).  Word, apparently, doesn't like PNG format.  We've redone Figure 2 with a clearer-looking font to match the others, which makes it less obvious.  The final plot should look better, since it won't have the issue with Word on top of the limited resolution.

---

## Author Response (AR2)

**Evaluation of Sub-Monthly Oceanographic Signal in GRACE "Daily" Swath Series Using Altimetry**

Jennifer A. Bonin[1] and Himanshu Save[2]

[1]College of Marine Science, University of South Florida, St. Petersburg, FL, 33701, USA
[2]Center for Space Research, University of Texas at Austin, Austin, TX, 78759, USA

*Correspondence to*: Jennifer A. Bonin (jbonin@mail.usf.edu)

*Greetings,*

*Thank you again to both reviewers as well as the editor, I hope the results are satisfactory to all.*

*Dr. Dobslaw's suggestions have all been accepted – except for the note that the spelling of "successful" vs "successfull" is apparently one of those American/British English differences!  Thank you again for your helpful review.*

*The anonymous reviewer's take on the MSS model bias change has also been taken into account.  I've taken the editor's advice, and moved that section to the supplemental material, where I agree that it fits better.  Thank you very much for that advice.*

*Please let me know if you have any other suggestions on how to improve this manuscript.*

*Jennifer Bonin*

[revised manuscript text omitted]

However, the transition from Jason-1 GDR-E to Jason-2 GDR-D creates a large, non-uniform bias (Fig. S1a). We have determined that this bias pattern comes from a change in the mean sea surface (MSS)
model used between the D and E versions of the GDR output. In version D, a 16-year MSS correction (MSS_CNES-CLS-2011) was created by averaging satellite altimetry data over the years 1993-2008. In version E, a different dynamic topography model (MSS_CNES-CLS-2015) was created over the longer

1993-2012 timespan instead. (For details on the Jason-1 GDR-E MSS, see the AVISO website: https://www.aviso.altimetry.fr/en/data/products/auxiliary-products/mss/mss-description.html. For details on the Jason-2 GDR-D MSS, see the Jason-2 Handbook: CNES et al., 2011.)

The change of MSS model results in two types of differences during the Jason-1 to Jason-2 leveling process. First, the newer GDR-E MSS model has a finer resolution and is averaged over differing years, which will result in slightly different values in some areas. We are currently assuming that any such differences will be small and will tend to cancel out when looking at data over the entire globe. The second, much more concerning issue is that of the mean bias. The MSS model used in GDR-E is referenced to a center date of 2003, while the GDR-D is referenced to a center date of 2001 – and no bias correction has been applied (or provided) to align the two averages to an identical epoch. In areas where the sea surface height is experiencing a trend, this will introduce an artificial jump between the two averages. Averages made from even identical input data would be offset from each other by a time-constant, but spatially-variable, bias of approximately the size shown in Figure 1a.

To correct for this properly, one would ideally need to reprocess both (or all three) sets of Jason data with a consistent MSS model. This being a lengthy process, however, we used the following approximate technique instead. We computed the average overlap bias (Fig. 1a) along the ground-tracks, then smoothed it with a 100km Gaussian smoother to remove very short-wavelength features, and used the value of that at each point as the Jason-2 bias correction. We note that it is very important to compute the value at each point, as there would be a ±4 cm spread in water heights otherwise, depending on where in the ocean you are looking. Using the mean bias over this map would result in a constant -0.785 cm correction, which would not be accurate in most places. (As a comparison, we found the Jason-2D to Jason-3D bias to be a larger but very stable -2.871 cm.)

Our ultimate results are double-differenced, comparing the statistics of Altimetry minus GRACE to Altimetry minus a model. Because of this, the treatment of the MSS model is non-critical. There could be an effect in the percent of altimetry's variance explained by each other series (Fig. 3) but that effect would cancel itself out in the difference in P.V.E. between two comparison series (Fig. 4). For this reason, the rough treatment we have used here is effective. However, any GDR user interested in looking at non-differenced results should be aware of this bias and correct for it by the consistent replacement of the MSS model.

[Figure]

**Figure S8:** Bias offsets between (a) Jason-1 GDR-E and Jason-2 GDR-D, and (b) Jason-2 GDR-D and Jason-3 GDR-D.

Moved (insertion) [1]

Page 4: [1] Deleted                    Jenn Bonin                    3/3/20 2:14:00 PM